# The MetaInvert soil invertebrate genome resource provides insights into below-ground biodiversity and evolution

Gemma Collins [1,2], Clément Schneider[2,3], Ljudevit Luka Boštjančić [1,4,5], Ulrich Burkhardt[6], Axel Christian[3], Peter Decker[3], Ingo Ebersberger [1,2,7], Karin Hohberg [3], Odile Lecompte [4], Dominik Merges[8], Hannah Muelbaier [7], Juliane Romahn[1,2], Jörg Römbke[9], Christelle Rutz[4], Rüdiger Schmelz[10], Alexandra Schmidt [1,11], Kathrin Theissinger[1,2,5], Robert Veres[1,12], Ricarda Lehmitz [3], Markus Pfenninger [1,2,13] & Miklós Bálint [1,2,14 ✉]

Soil invertebrates are among the least understood metazoans on Earth. Thus far, the lack of taxonomically broad and dense genomic resources has made it hard to thoroughly investigate their evolution and ecology. With MetaInvert we provide draft genome assemblies for 232 soil invertebrate species, representing 14 common groups and 94 families. We show that this data substantially extends the taxonomic scope of DNA- or RNA-based taxonomic identification. Moreover, we confirm that theories of genome evolution cannot be generalised across evolutionarily distinct invertebrate groups. The soil invertebrate genomes presented here will support the management of soil biodiversity through molecular monitoring of community composition and function, and the discovery of evolutionary adaptations to the challenges of soil conditions.

[1] Senckenberg Biodiversity and Climate Research Centre, Frankfurt am Main, Germany. [2] LOEWE Centre for Translational Biodiversity Genomics, Frankfurt am Main, Germany. [3] Soil Zoology, Senckenberg Museum of Natural History, Görlitz, Germany. [4] Department of Computer Science, ICube, UMR 7357, University of Strasbourg, CNRS, Centre de Recherche en Biomédecine de Strasbourg, Strasbourg, France. [5] Department of Molecular Ecology, Institute for Environmental Sciences, Rhineland-Palatinate Technical University Kaiserslautern Landau, Landau, Germany. [6] Soil Organism Research, Görlitz, Germany. [7] Institute of Cell Biology and Neuroscience, Goethe University, Frankfurt am Main, Germany. [8] Department of Forest Mycology and Plant Pathology, Swedish University of Agricultural Sciences, Uppsala, Sweden. [9] ECT Oekotoxikologie GmbH, Flörsheim, Germany. [10] Freelance Biologist, A Coruña, Spain. [11] Limnological Institute, University of Konstanz, Konstanz, Germany. [12] Institute of Biology and Geology, Babeș-Bolyai University, Cluj-Napoca, Romania. [13] Johannes Gutenberg University, Mainz, Germany. [14] Department of Insect Biotechnology, Justus-Liebig University, Gießen, Germany. ✉email: miklos.balint@senckenberg.de

Soils and soil biodiversity are becoming increasingly valued and protected at the policy level[1]. Soil invertebrates are major components of soil biodiversity, and their activity is important for almost all soil ecosystem services[2]. For example, soil invertebrates are responsible for up to 50% of the litter decomposition[3]. They contribute to functional services crucial to humans, such as nutrient cycling, water storage and support above-ground food production through the integration of nutrients in food webs[4–6]. Furthermore, soil invertebrates play major roles in regulating microbial activity along the plant-soil continuum[7]. Consistent with their importance in soil ecosystems, they are actively promoted in soil biodiversity conservation frameworks[8].

However, soil invertebrates are inherently difficult to study morphologically due to their incredible diversity, huge abundances, and small body size with microscopic morphological details. Though generally tiny, they show a ~100-fold variation in body weight, which ranges from nanograms to grams[9]. There are potentially hundreds of thousands of undescribed species globally[10]. Moreover, taxonomic expertise is declining[11] and this is particularly problematic for groups where experts have always been rare.

DNA- and RNA-based methods are long promoted to support traditional taxonomy and ecological studies in difficult organism groups. Shotgun metagenomics randomly sequences DNA fragments from a sample, instead of relying on PCR-amplified taxonomic marker genes. Metagenomics is an increasingly feasible approach to record the presence of higher eukaryotes in a diverse range of samples[12–14]. Since metagenomics can utilise all genomic information for taxonomic identification, it has improved sensitivity and specificity compared to metabarcoding[15], and it promises superior quantification of species' biomass[16]. Metatranscriptomics in turn records genes which are actively transcribed into RNA and thus drive ongoing biological processes[17], informing about the metabolic activity of soil community members[18], and functional changes in these communities[17].

Comprehensive genome collections are the backbone for metagenomics and metatranscriptomics. If genome databases are available, shotgun metagenomics and metatranscriptomics have shown to provide unprecedented insights[17,19], e.g., into vegetation change over glacial cycles[15], historic population genomic processes[20,21], and kingdom-spanning processes of ecosystem functioning[22]. Large genome sequencing initiatives like the Earth Biogenome Project[23] will provide this data ultimately, but progress so far mainly focused on large, prominent organisms, such as mammals[24], birds[25], insects[26] and plants[15]. In addition to serving taxonomic identification, broad (many distinct groups) and dense (many species from a group) sequencing of genomes additionally allows identifying common patterns of gene evolution and test the taxonomic generality of hypotheses on genome evolution.

## Results and discussion

**A genome resource for soil invertebrates.** Here, we have generated a large genomic resource to support insights into the structure, activity and functioning of soil invertebrate communities (Fig. 1). We had two aims. First, we wanted to provide a large number of soil invertebrate genomes to aid species identifications through metagenomics or metatranscriptomics. Second, we intended to explore patterns of genome evolution across taxa, which needs a taxonomically broad and dense sampling of genomes. We sequenced and assembled the genomes of 232 species, representing 14 common soil invertebrate groups (hereafter referred to as "groups") encompassing 94 families, most of which were lacking whole-genome data thus far (Fig. 2, Table 1), including Collembola ($n = 87$ species), Oribatida ($n = 62$), two

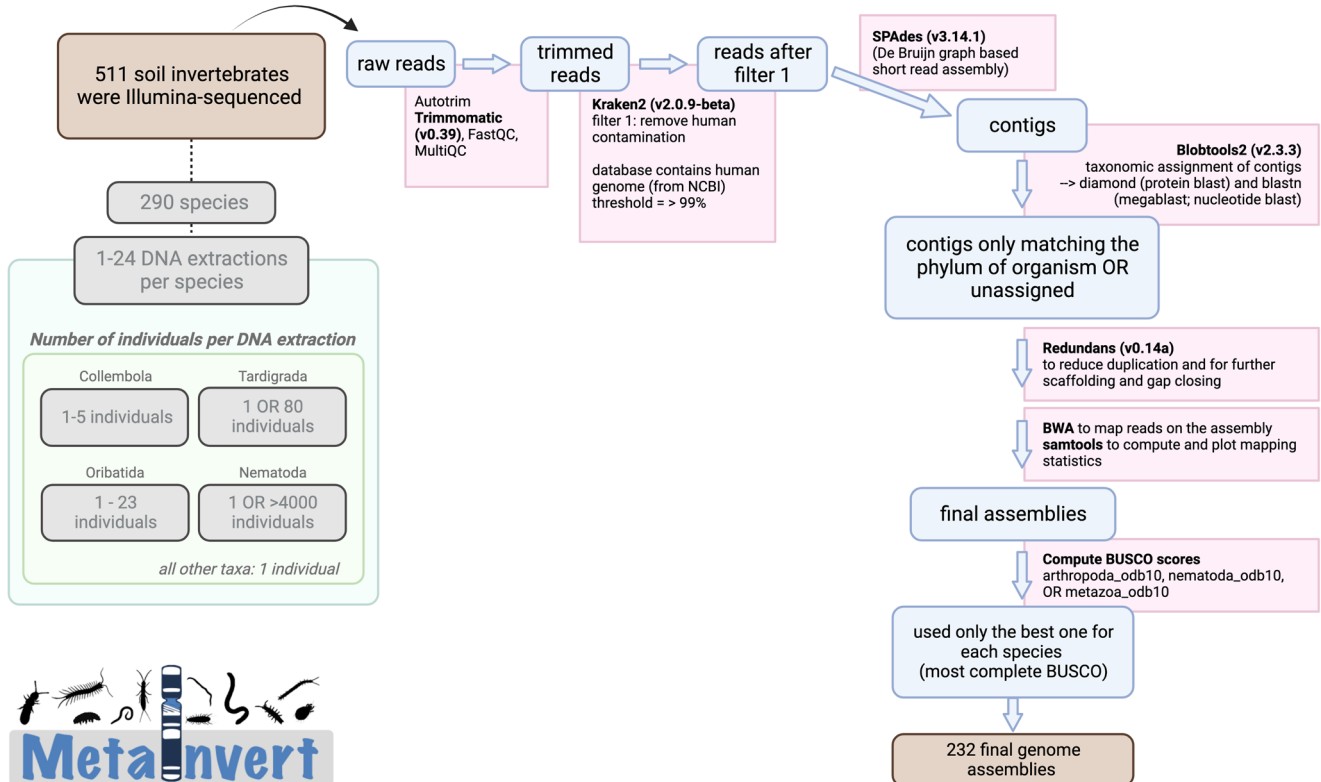

**Fig. 1 Overview of the bioinformatic pipeline for genome assembly and quality control.** The genome assembly pipeline consists of a read quality filtering step, short read assembly and several steps for removing non-target DNA reads, co-sequenced along the genomes of the targeted species. The MetaInvert logo was created by the first author. Animal silhouettes originate from phylopic.org, and they can be reused under Creative Common licences.

classes of Myriapoda ($n = 23$ Diplopoda; $n = 19$ Chilopoda) and Nematoda ($n = 18$). Genome completeness estimated with benchmarking universal single copy orthologs (BUSCO)[27] was 59.78% on average (median: 69.2%), with an average contig N50 of 6080 bases (median 4039), and with an average L50 of 28,375 (median: 11503, Supplementary Data 1).

**Improved taxonomic assignment of metazoan environmental sequence data.** To demonstrate the relevance of this genomic resource, we first used the 232 genomes to improve the taxonomic assignment metatranscriptomic sequences generated from a 2-year sampling of soil environmental RNA (eRNA) along an

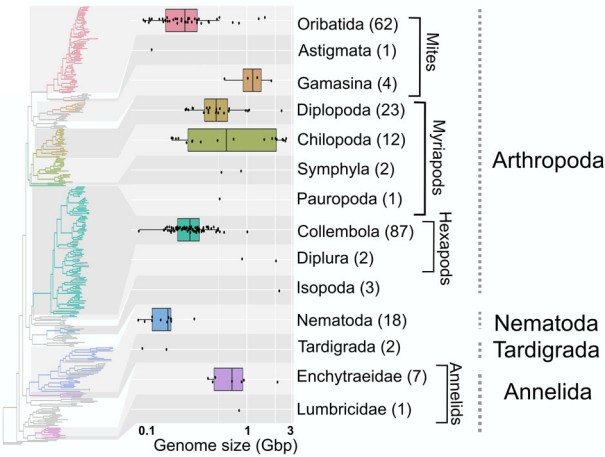

**Fig. 2 Maximum likelihood phylogenetic tree of soil invertebrate genomes.** The tree is based on an alignment of 141 metazoan BUSCO genes of the 232 soil invertebrates sequenced in this work (coloured branches), and 118 NCBI RefSeq (grey branches), representing four phyla. A high-resolution, annotated version of the tree is available as Supplementary Fig. 1. A more detailed tree and the alignment are available on FigShare[30]. Boxplots reflect the genome size distribution of the taxa subsumed in the corresponding clades in gigabases (Gb). Numbers of sequenced genomes with genome size estimates are indicated for each group. Genome size estimation was not possible for some of the assemblies. Genome size estimates can be found in Supplementary Data 1. Center line: median; box limits: upper and lower quartiles; whiskers: 1.5× interquartile range; points: outliers.

elevational gradient[28]. Such assignments of soil eRNA were previously limited in scope due to a general lack of soil invertebrate genome data. Briefly, we assigned eRNA reads with bacterial, fungal, plant and soil invertebrate genomes, with and without including the MetaInvert genomes presented here. We found that about 2.45% (854,409 reads) of the classified metatranscriptomic reads (40,265,768) could be assigned to soil invertebrates, in comparison to bacteria (77,1%, 31,063,088), fungi (20.1%, 8,078,679), and plants (0.33%, 134,852)[29]. Previous metatranscriptomic studies reported a similar microbial eukaryote to bacteria ratio[29,30]. The inclusion of the MetaInvert genomes significantly increased reads assigned to soil invertebrates (Kruskal-Wallis $X^2 = 9.14$, $df = 1$, $p = 0.002$, Fig. 3a). We recorded 11 soil invertebrate classes (Fig. 3b), of which the most abundant were nematodes of the class Chromadorea followed by clitellates (comprising both earthworms and enchytraeids). Linear regression showed a marked dip in soil invertebrate richness along the elevation gradient (ANOVA, $F_{elevation} = 0.22$, $p_{elevation} = 0.65$, $F_{elevation^2} = 9.1$, $p_{elevation^2} = 0.02$, Fig. 3c). This is in contrast with findings of hump-shaped elevation - richness relationships in soil invertebrates[31]. The pattern observed by us might be driven by distinct vegetation covers, although the confirmation of this needs a better sampling resolution. The community composition of soil invertebrates showed no statistically significant changes along the elevation gradient (analysis of deviance of multivariate generalised linear model fits, $df = 8$, $dev = 434.60$, $p = 0.13$), marginally significant differences across habitats study years ($df = 65$, $dev = 806.03929.87$, $p = 0.085$), and statistically significant differences between the two study years ($df = 5$, $dev = 1066.09$, $p = 0.04$).

No change in community composition along the elevation gradient is consistent with observed high abundances of soil invertebrates at high altitudes[30]. Differences in vegetation are known to influence soil invertebrate community composition, although our analysis may lack power to equivocally detect these. Differences in community composition between the study years may reflect year-specific environmental differences. However, we caution not to over interpret these results. The power of an analysis of drivers of community composition and richness on this gradient should be increased with more extensive sampling. The analyses nonetheless demonstrate the value of a dedicated soil invertebrate genome database for the identification of shotgun-sequenced environmental nucleotide samples from soils.

**Table 1 Overview of 232 soil invertebrate genome assemblies.**

| Phylum | Taxon group [rank] | Common name | n known species (soil or terrrestrial) | n species (published genomes) | n species (genomes contributed here) |
|---|---|---|---|---|---|
| Annelida | Lumbricidae [family] | Earthworms | 7000 | 2 | 1 |
| Annelida | Enchytraeidae [family] | Potworms | 700 | 1 | 7 |
| Nematoda | Nematoda [phylum] | Nematodes | 25000 | 73 | 18 |
| Tardigrada | Tardigrada [phylum] | Tardigrades | 1150 | 4 | 2 |
| Arthropoda | Gamasina [infraorder] | Predatory mites | 40000 | 1 | 4 |
| Arthropoda | Astigmata [suborder] | Mites [not soil] | | 7 | 1 |
| Arthropoda | Oribatida [suborder] | Box mites | | 7 | 62 |
| Arthropoda | Chilopoda [class] | Centipedes | 3000 | 2 | 19 |
| Arthropoda | Diplopoda [class] | Millipedes | 12000 | 3 | 23 |
| Arthropoda | Symphyla [class] | Symphylans | 200 | 0 | 2 |
| Arthropoda | Pauropoda [class] | Pauropods | 800 | 0 | 1 |
| Arthropoda | Isopoda [order] | Pill bugs | 3637 | 5 | 3 |
| Arthropoda | Diplura [order] | Diplurans | 1000 | 2 | 2 |
| Arthropoda | Collembola [class] | Springtails | 8500 | 35 | 87 |

For each taxonomic group we also list the number of species with publicly available genome assemblies (as of June 2022).

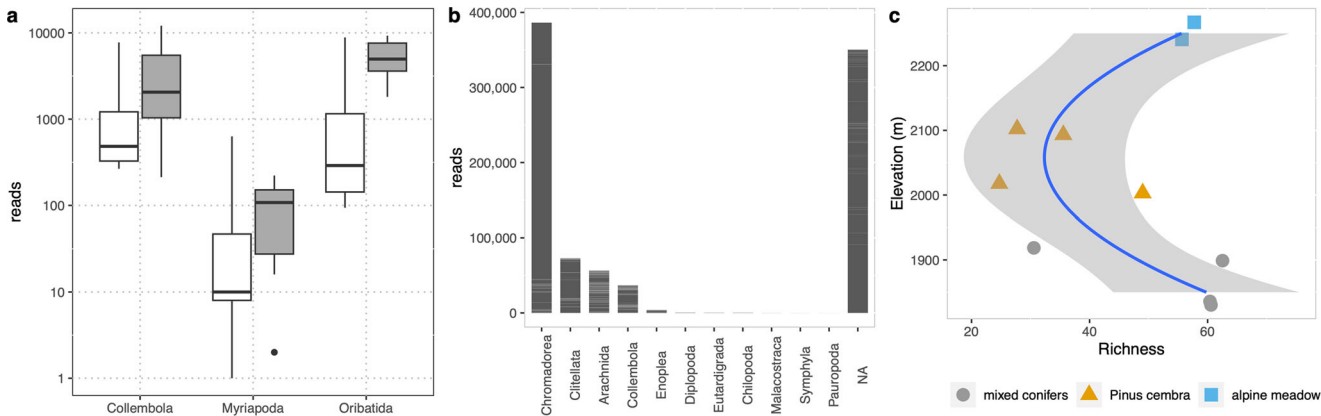

**Fig. 3 Taxonomic assignments of soil metatranscriptomes using soil invertebrate genomes. a** Assignment success of soil metatranscriptomic reads using genomes available in NCBI RefSeq (white) and MetaInvert genomes in addition to NCBI genomes (grey). Please note the log-scale of the y-axis (taxon observations with RefSeq genomes: Collembola $n = 290$, Myriapoda $n = 80$, Oribatida $n = 90$; independent taxon observations with RefSeq + MetaInvert genomes: Collembola $n = 810$, Myriapoda $n = 270$, Oribatida $n = 630$), center line: median; box limits: upper and lower quartiles; whiskers: 1.5× interquartile range; points: outliers; **b** reads assigned to common soil invertebrate classes, with NA marking metazoan reads not assigned to soil invertebrates at the class level; **c** soil invertebrate richness trend along an elevation gradient (grey area marks standard error of the trendline). Assignments are available as Supplementary Data 2.

**Insights into genome size evolution.** As a second example, we addressed hypotheses concerning genome size evolution. We estimated the genome size for 191 species using the assembly-based approach ModEst[31]. We found a 30-fold range of genome sizes across the groups (Fig. 2), from 79 Mb (the nematode *Discolaimus major*) to 2.9 Gb (the chilopod *Lithobius crassipesoides*). Nematoda and Tardigrada had typically small genomes, whereas the genomes of Enchytraeidae were remarkably larger. In addition to between-group variation, some groups also had a wide range of genome sizes among member species. For example, Chilopoda (centipedes) genomes ranged in size from 0.178 to 2.90 Gb, while Oribatida genomes ranged from 0.09 to 1.72 Gb. Repeat content and GC content also varied widely both within and between soil invertebrate groups (Supplementary Fig. 2).

Classic theory predicts that a few basic factors, in particular effective population size, should lead to causal relationships between genome properties and functional traits (Fig. 4a)[32–34]. However, recent studies have shown that taxon-specific processes might be more important for genome size than demography[35,36]. We used our taxonomically broad data set to test the classical hypothesis of a few factors generally influencing genome size evolution vs. a more lineage-specific view with a series of structural equation models (SEMs, Fig. 4). We used genomes with at least 50% BUSCO completeness and 8× mode coverage. To parametrize the SEM and connect the 143 new genome assemblies with ecological traits, we first gathered trait data from original literature. Information about habitat preferences was added from the Edaphobase data warehouse for soil biodiversity (https://portal.edaphobase.org/). We focussed on three traits: (a) body length as a proxy for body size (minimum female adult body length for nematodes, and mean adult body length for all other taxa), (b) reproduction mode, and c) the number of known habitat types where a species occurs, as a proxy of habitat generality (based on CORINE—Coordination of Information on the Environment[37]). We annotated repetitive elements with species-specific repeat libraries. We estimated effective population size (theta) directly from the genome data by making use of the genome-wide heterozygosity in the reference individual. This proxy measure of effective population size was calculated individually for each genome assembly with at least 8X coverage. Genomic and ecological traits are accessible in Supplementary Data 1.

The variables tested have complex interactions that need to be modelled in the SEMs. Effective population size should be influenced by habitat generalism, with the expectation that species able to thrive in a wide range of habitats should have larger population sizes and therefore also larger effective population sizes ($N_e$)[38]. $N_e$ should be inversely related to body size, as larger populations of small-bodied organisms can be maintained by the same amount of resources in comparison to large-bodied species[32]. The reproductive mode is known to impact $N_e$, because the higher the degree of inbreeding, the smaller the expected $N_e$[39]. High $N_e$ is frequently hypothesised to contribute to reducing repeats as evolutionary burdens from genomes, as selection is more efficient in larger populations[33,34]. Repeats are frequently considered to increase genome size[40,41]. If the repeats themselves are biased in base composition, this should reflect in the overall GC content. Interestingly, GC content is also linked to resource availability[42], which may be linked to habitat generalism via higher metabolic flexibility[43]. Even though most of these observations originate from bacterial studies, ample evidence exists that the environment may influence base composition also in metazoans[42,44–46].

When modelling all soil invertebrate groups together, most hypothesised causal relationships were either statistically insignificant or pointed to the opposite directions than classical theory predicted (Supplementary Fig. 3). Most strikingly, high $N_e$ size was linked to higher repeat content which in turn implies larger genome size. This suggests that efficient selection does not universally reduce the evolutionary burden of large genomes and repeat content[47]. The SEMs supported only two of the hypothesised causal relationships when these were modelled for all taxa together (Fig. 4b): a positive link between repeat content and genome size, and a negative link between repeat content and GC content. Genome size is frequently considered to be driven by repeat content[48,49], but with variation in the relationship among higher taxa of vertebrates[50]. Such variation might be due to epigenetic regulation via repetitive elements, maintenance of chromosome structure[51], and modification of gene expression and transcript diversification[52]. Higher GC content is linked to smaller genome size in many but not all eukaryotic groups[49]. This link might also originate from the expansion of repeats with low GC content.

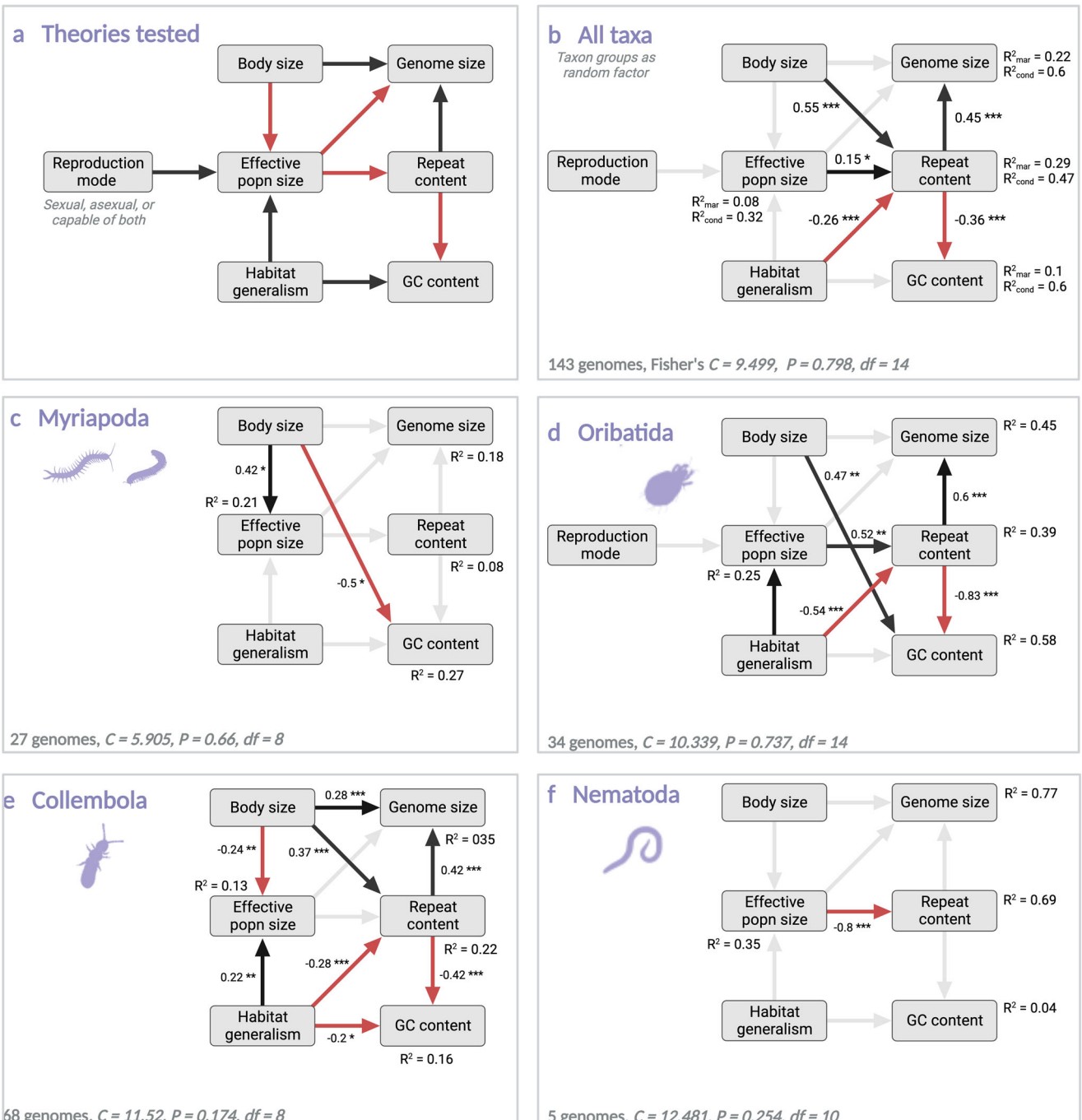

**Fig. 4 Structural equation models (SEMs) of hypothesised causal relationships among genomic traits and their ecological drivers. a** Initial SEM with hypothesised links; **b–f** SEMs fitted to all taxa, and to major taxonomic groups. Arrows indicate hypothesised or modelled relationships, positive (black) or negative (red). Links marked with grey arrows were not statistically significant in the SEM. Fisher's C evaluates conditional independence claims among nodes and indicates model fit, with p-values showing whether discrepancies between the model and the data are statistically significant. Degrees of freedom are marked with df. Values next to arrows show standardised estimates, with asterisk indicating the statistical significance of the relationship (*p < 0.05, **p < 0.01, ***p < 0.001). Animal silhouettes originate from phylopic.org, and they can be reused under Creative Common licences.

Our results confirm that the strength and direction of relationships among genome size, repeat content, GC content, and their ecological drivers vary among higher taxa of invertebrates (Supplementary Fig. 3). SEMs fitted separately to higher taxa (myriapods, oribatid mites, springtails, nematodes) showed marked group-specific differences in the support of causal hypotheses between genomic and ecological traits (Fig. 4c–f, Supplementary Fig. 3). The assumed positive link between body size and genome size[32] received statistical support only in Collembola, but with an opposite sign as predicted by the nucleotypic theory[32]. The effects of body size on genomes are often difficult to disentangle from other co-variables[53,54]. This indicates lineage-specific expansion or contraction of genomes, reported for diverse eukaryotes[50,55]. The expected negative relationship between $N_e$ and repeat content[40,41] was confirmed only in nematodes. However, the relationship was positive in oribatid mites, and missing altogether in the other taxa. Habitat generalism was positively linked to effective population size only

in Collembola and oribatids, but not in myriapods and nematodes. This suggests that generalists may not be as fit as specialists in any particular habitats[56,57], and their evolution might depend on differential rates of population evolution compared to rates of environmental change[58]. Interestingly, models of oribatids and Collembola suggested that higher habitat generality might be linked to lower repeat contents. Altogether, our analysis supports a more nuanced, lineage-specific view of factors driving genome size evolution rather than the classical view of only a few general factors governing the C-value enigma.

**Gene loss patterns in springtails and oribatid mites**. As a third example, we explored whether shared gene loss might be related to repeated adaptations of phylogenetically distant metazoans to soil conditions. Gene loss is a key process in evolution[59,60]. Here, the dense taxon sampling for individual groups allowed to differentiate between consistent gene absence across several taxa, which likely indicates gene loss, and the sporadic absence of a gene in individual taxa, which likely represents noise introduced by assembly incompleteness. To further reduce the risk that incomplete gene annotations generate a spurious signal of gene loss, we used a targeted search for orthologs in the un-annotated genome assemblies to determine the presence/absence patterns of genes across taxa. We analysed the presence of 1482 core metazoan gene orthologs. Notably, this revealed that 50 core genes are missing in springtails ($n = 78$ species), and 97 core genes were not found in the oribatid mites ($n = 54$ species) (Fig. 5). Given the large number of investigated taxa in the two groups, it is unlikely that these genes have been accidentally missed. Instead, their absence indicates gene losses early during diversification of the respective groups, similar to what has been seen for other animals[61]. Overall, fifteen gene ontology terms were significantly enriched (testFisher < 0.05) among the genes lost involving biological processes such as tubulin metabolism and cellular and subcellular movement (Oribatida). There was a significant loss of genes involved in pyridine-containing compound metabolic processes in springtails (Fig. 5; Supplementary Data 3, 4). Pyridine-containing molecules have a considerable spectrum of antimicrobial and antiviral activities[62], and associated gene loss might be related to the gain of endogenous antibiotic synthesis ability by many springtail species[63]. We also manually screened the UniProt database (accessed on 28.6.2023) for putative gene functions associated with genes missing from Collembola and Oribatida assemblies. We aimed to identify functional or other relevant commonalities among the genes which might be missed by an algorithmic GO enrichment analysis. We could not detect patterns in gene functions. It was noteworthy that all existing annotations originated from only two species: *Drosophila melanogaster* or *Strigamia maritima*. This highlights the general difficulties with transferring annotations gained from a few model taxa to the breadth of biodiversity, with targeted annotation of specific genes being a solution.

In summary, our large collection of soil invertebrate genomes is a first major step towards a comprehensive DNA- or RNA-based identification of the entire soil biodiversity: they extend the scope of metagenomic or metatranscriptomic studies from microorganisms to metazoans. An important limitation of the study is the quality of the genomes, which precludes deeper analyses, such as structural comparisons. Genome quality is currently restrained by the qualitative and quantitative requirements of the current sequencing techniques with respect to genomic DNA. Although it is already possible to generate highly contiguous and complete genomes of soil invertebrates from single specimens[64], the minute amounts of genomic DNA (often fragmented because of field preservation) does not yet allow for the generation of better

quality genomes on scale. Nonetheless, the genomes are of sufficiently high contiguity or completeness to considerably improve metagenomic and metatranscriptomic sequence assignments[65]. Further, the taxonomically broad and dense sampling of genomes provides unique insights into genome evolution, although clearly not into structural differences. Here we could show that no single theory of genome evolution fits all taxa: there are probably no simple overarching explanations for observed variations in genome properties, but interactions of multiple drivers result in divergent genome evolution patterns in different groups, reflecting their unique evolutionary history. Broad genome sampling allows for the identification of group-specific gene loss patterns, highlighting issues and future directions around the functional annotation of genomes from non-model taxa in diverse habitats. Overall, the 232 soil invertebrate genomes demonstrate the importance of genome sequencing efforts for understanding the ecology and evolution of the full scale of eukaryotic biodiversity, and project a future when maximum taxonomic and functional information will be gained from every environmental DNA or RNA fragment.

## Methods

**Specimen sampling and species-level identification**. Specimens were collected in the field or obtained from cultures, supplemented with existing soil invertebrate specimens from Senckenberg museum collections (Supplementary Data 1). Sampling occurred between 2011 and 2020, mostly in Germany, but in some cases also from countries in Europe. Soil macrofauna was mainly collected by hand, whereas meso- and microfauna were obtained from soil samples with MacFadyen[66] or Baermann extraction[67]. DNA was extracted from over 500 single specimens, or occasionally from multiple individuals (single-species cultures of tardigrades and smaller-bodied nematodes, Supplementary Data 5). A non-destructive DNA extraction method[68] was preferred and used where possible. Otherwise, the MagAttract High Molecular Weight DNA Kit (Qiagen, Hilden, Germany) was used, mostly for cultured specimens. Voucher specimens are deposited in the Senckenberg museum collection in Görlitz.

For larger taxa such as Chilopoda, Diplopoda, Isopoda, Enchytraeidae and Lumbricidae, the species-level morphological identification was possible before DNA extraction, and only a single leg, a few body segments or musculature of mouthparts were used for DNA extraction, the rest of the body was kept as a voucher. For medium sized taxa like Acari and Collembola that normally would require clearing in lactic acid prior to species identification, the specimens were presorted on family or genus level, the whole specimens were used for non-destructive DNA extraction, and finally species-level identifications were carried out with recovered vouchers. In cases where non-destructive DNA extraction did not deliver sufficient amounts of DNA or the voucher was lost during extraction, identification was validated by aligning species markers (28 S, COI) from the whole-genome sequence data with existing species markers in GenBank or generated by us. For small, soft-skinned taxa (Nematoda, Tardigrada), where non-destructive DNA extraction is not possible, two different sources/techniques were used: (1) for most species, specimens were derived from own established cultures with known taxon and strain names, or (2) where such cultures did not exist, we freshly Baermann-extracted specimens from soil samples and identified morphospecies with at least 6 specimens at 400x magnification under an inverted microscope. We then extracted DNA from half of the specimens and prepared permanent slides of the other half (vouchers). We assigned species identity to the genome-sequenced specimens, if all vouchers were identified as the same species[69].

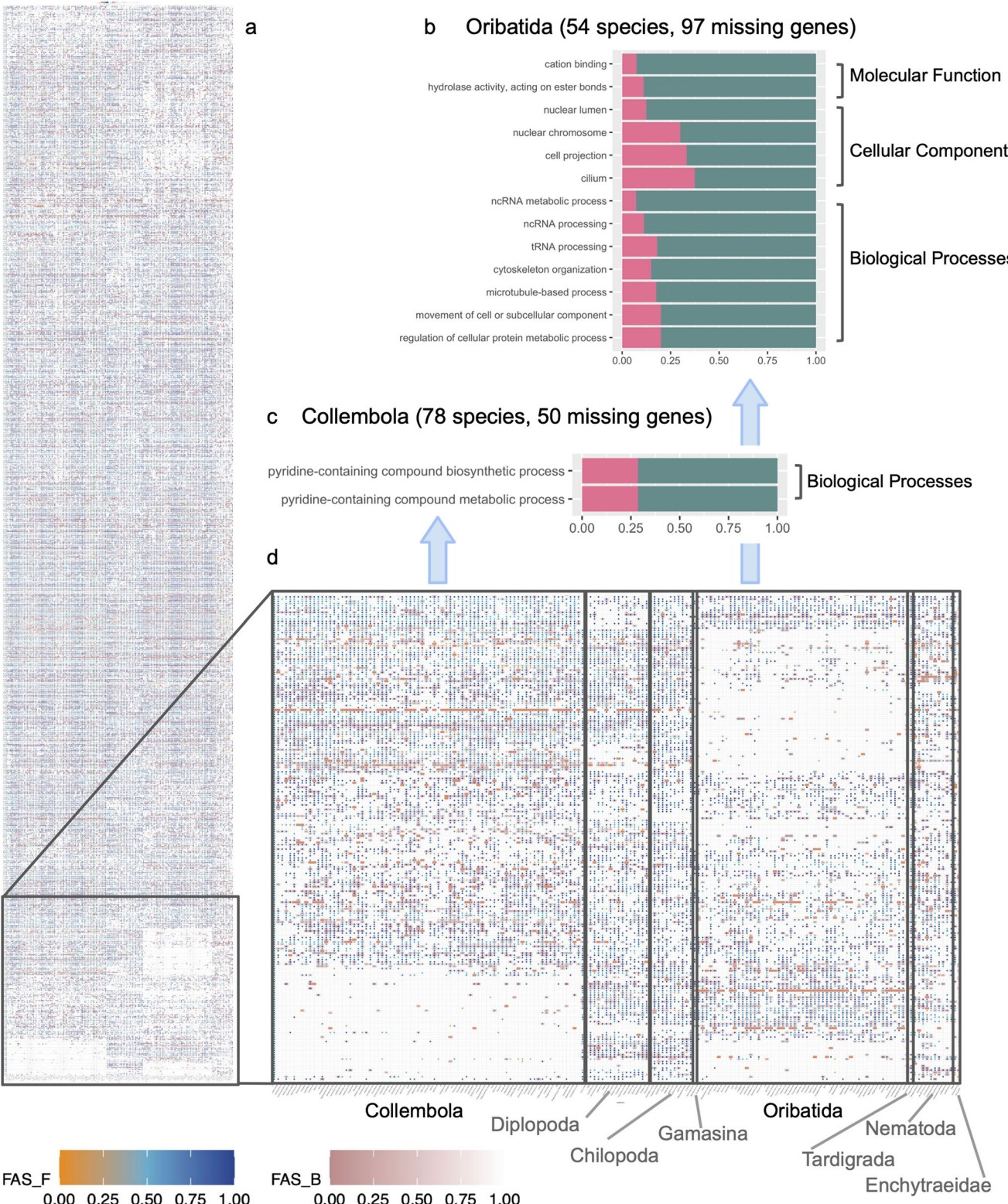

**Fig. 5 Loss of metazoan core genes in soil invertebrate species.** phylogenetic profiles of 1482 metazoan core genes across 177 soil invertebrate species; fraction of genes annotated with GO terms in the loss set (red) and in the background set (green) in oribatid mites **b** and springtails **c**; **d** genes consistently missing in springtails or oribatid mites. Colours in **a**, **d** represent feature architecture similarity among the identified orthologs and the reference gene, with a score between 1 (same architecture) and 0 (dissimilar architecture, or no features in the reference protein). The score is computed once by comparing the reference gene with the identified ortholog (FAS_F, dots on the graphic), and once by comparing the identified ortholog with the reference gene (FAS_B, background colour to dots). Data underlying the GO enrichment analysis are available as Supplementary Data 4.

**Illumina sequencing.** Sequencing libraries for each specimen, or pool of specimens, were prepared in-house at Senckenberg, Frankfurt, Germany with the BEST protocol[70] or with the NEBnext ULTRA II DNA Library Prep Kit, according to the manufacturer's protocol. Short-read Illumina sequencing (300-bp paired-end) was carried out at Novogene Europe (Cambridge, UK) using the NovaSeq 6000 platform, with unique dual indexing as the library tagging strategy for multiplexing on the lanes. Our central aim with the genome database was to improve species identifications. As this can be achieved with low sequencing coverage[65], our initial sequencing efforts targeted 2 gigabase (Gb) per species. We increased efforts to 10 Gb per species as sequencing became more affordable. For most of the reported genomes we obtained ~10 Gb per species.

**Genome assembly pipeline.** We established a pipeline to assemble reads into draft genomes (Fig. 1). First, the sequencing adapters were trimmed using Trimmomatic (v0.39; paramaters: ILLUMINACLIP:adapters.fasta:2:30:10:8:true SLIDINGWINDOW:4:20 MINLEN:50 TOPHRED33[71]). The trimmed reads were queried against the human genome (GRCh38 assembly on NCBI) using Kraken2 (v2.0.9-beta; --confidence set to 0.2, other parameters default[72]), and all 'human' positive reads, if any, were discarded. The remaining reads were then assembled using SPAdes (v3.14.1; default settings[73]). The resulting contigs were then queried against the NCBI non-redundant nucleotide database using blastn (megablast mode, -max_target_seqs 10, -max_hsps 1, -evalue 1e-25), and against the NCBI non-redundant protein database using Diamond (blastx mode, --sensitive --max-target-seqs 1, --evalue 1e-25[74]). NCBI databases were downloaded on 27-Oct-2020. Blobtools2 (v2.3.3[7]) was used to perform a taxonomic assignment based on the Blast and Diamond results, using the 'bestsumorder' rule. The contigs assigned to the phylum of the target organism as well as the unassigned contigs were kept (i.e., contigs assigned to other phyla were considered obvious contaminants and removed). Redundans (v0.14a[75]) was used to reduce the amount of duplication in the retained contigs, as well as further scaffolding and gap closing (default parameters were used). The resulting scaffolds were used as the final assembly draft for subsequent analyses. The Burrows-Wheeler Aligner (BWA) was then used to map the reads on the assembly and samtools[76] (v1.11-2-g26d7c73) to compute and plot the mapping statistics (e.g., GC content).

**Quality assessment of assemblies using BUSCO.** Benchmarking Universal Single-Copy Orthologs (BUSCO) databases[27] are sets of genes for specific taxon groups, where every gene in the BUSCO set is expected to be present once in each member species. We searched for BUSCO genes in our final assemblies as a quality indicator of genome assembly completeness, we used the most specific BUSCO database that was available for each of the invertebrate groups (nematoda_odb10 BUSCO genes for nematode assemblies, arthropoda_odb10 for arthropods, metazoa_odb10 for tardigrades, enchytraeids and earthworms). We selected the genome assembly with the highest percentage of complete BUSCO genes as the species representative if more than a single replicate per species was available. This resulted in a total of 232 genome assembly drafts used for downstream analyses.

**Improving metatranscriptomic assignments.** Metatranscriptomic reads were generated from soil samples collected along an elevation gradient spanning 400 m of elevation in the Alps[28,77,78]. Briefly, short soil cores were taken and preserved in LifeGuard (Qiagen, Hilden, Germany) in 2015 and 2017. RNA was extracted with an RNeasy PowerSoil Total RNA Kit (Qiagen) from ten cores. RNA sequencing libraries were prepared of each

RNA extracts with a NEBNext Ultra RNA Library Prep Kit (Frankfurt am Main, Germany), and 8 gigabases of each library were sequenced at Novogene (UK) on an Illumina NovaSeq6000 sequencer in a 150 bp paired-end reaction. Reads were trimmed of adapters with Trimmomatic[71]. Reads were taxonomically assigned with kraken2[72] in a three-step process. First, we screened the metatranscriptomes against the human genome for eventual human contamination. Second, we assigned remaining reads with a custom database containing all bacterial, plant and fungal reference genomes from NCBI (accessed on 15.1.2023). Third, we then tested the impact of a dedicated genome database for soil invertebrate detection: unassigned reads from the second step were mapped against all springtail (57), oribatid mite (9) and myriapod genomes (8) available in NCBI RefSeq as of 20.6.2023, with and without including the 232 MetaInvert genomes (Supplementary Data 2). We visualised the richness of soil invertebrates along the elevation gradient at the genus level. As nucleotide sequence counts are not normally distributed and they are frequently overdispersed[79], we evaluated differences in community composition among the study years and habitats, and along the elevation with a model-based analysis of multivariate abundance data[80]. Community analyses were performed in R v4.2.2[81].

**Building the phylogeny using metazoan BUSCO genes.** We searched for BUSCO genes with the metazoan_obd10 database (v4.1.4) to generate a single phylogeny of the 232 soil invertebrate genomes and a selection of 118 publicly available invertebrate RefSeq genomes from NCBI (downloaded on 16.09.2021). The RefSeq genomes were included if they a) were from the same taxon group as our specimens, b) served to shorten the evolutionary distance between taxa in the tree. More specifically, we included any chromosome-level Protostomia genomes (excluding Insecta), genomes of any assembly quality for species within our 14 taxonomic groups of interest, and some additional specific outgroups (two Echinodermata, three Rotifera, a Priapulida, *Machilis hrabei* and *Drosophila albomicans*). We found 141 metazoan BUSCO genes which were present in at least 75% of the genome assemblies (Supplementary Data 1, 6). The phylogenetic approach is based on the https://github.com/mag-wolf/BUSCO-to-Phylogeny pipeline. We aligned these with Mafft (v7.481[82]) with 1000 iterative refinements. These gene alignments were then concatenated into a supermatrix using FASCONCAT (v1.04[83]) and trimmed using clipkit (v1.1.5[84]), keeping only parsimony-informative and completely conserved sites. We used IQ-TREE (v2.0.3[85]) to build four separate maximum likelihood trees (each with 1000 bootstrap replicates), selecting the best one based on the -log Likelihood value closest to zero[86]. We used R to visualise the phylogeny, using the packages ggtree (v3.1.5.900[87]), tidyverse, treeio (v1.17.2[88]) and colorspace[89].

We note the placement of Tardigrada in our phylogeny is next to Nematoda which is in disagreement with the currently accepted view that they should be closer to Arthropoda[90]. This is likely an artefact due to lack of public outgroup data[91,92], and has no downstream consequences for our analyses.

**Estimating genome size.** To estimate genome size, we used ModEst[31] which yields results comparable in accuracy to flow cytometry, the main non-sequencing method of genome size estimation, even from incomplete genomes. Briefly, we first plotted the distribution of sequencing coverage across each genome and visually inspected each plot for the mode coverage (the highest point of the peak). If a genome assembly did not have a clearly discernible peak in sequencing coverage then genome size was not estimated for this species. Otherwise, genome size was

estimated by dividing the total mapped bases by the mode coverage.

**Estimating effective population size.** Using mlRho (v2.9) we estimated theta directly from the genome data by making use of the genome-wide heterozygosity in the reference individual. This proxy measure of effective population size was calculated individually for each genome assembly with at least 8X coverage, twice as high as recommended by Haubold et al.[93].

**Annotating repeat content.** In addition to investigating several genome properties (i.e., GC content, BUSCO gene content, genome size and effective population size), and because repeat content is particularly relevant for explaining genome size variation among species, we also annotated the repetitive elements. Species-specific repeat libraries were constructed using the automated RepeatModeler (v2.0.1) pipeline with LTR Structural discovery pipeline activated[94]. For each genome, the resulting repeat libraries were merged with the RepBase (v26.05) Arthropoda-specific section[95] and subsequently used for the annotation and estimation of proportion of repetitive elements with Repeat-Masker (v4.1.2-P1[96]).

**Ecological trait data.** To connect the 232 new genome assemblies with ecological traits of the respective species, we first gathered existing functional trait data from Edaphobase (https://portal.edaphobase.org/) and from literature. We focussed on a) body length (minimum female adult body length for nematodes, and mean body length for all other taxa) as a proxy for body size, b) reproduction mode, and c) known occurrences in different soil habitat types (based on level 2 hierarchies described by the Coordination of information on the environment (CORINE)[37]). We provide this collected information as an additional database resource in Supplementary Data 1.

**Structural equation models.** We tested established or hypothesised causal relations among genomic, life-cycle and ecological variables through a series of structural equation models, with the aim of resolving multivariate relationships from the many interrelated variables. We selected only genomes with at least 50% BUSCO completeness and 8X mode coverage. Log transformations were applied to body size variables (due to non-normal distribution as determined by a two-sided Kolmogorov-Smirnov test ($p < 0.01$). We fitted the SEMs with piecewiseSEM (v2.1.0[97]). We performed the path analyses for all taxa together (linear mixed effect models, with soil invertebrate groups as random variable), and separately for each of the more densely sampled taxa (Collembola, Oribatida, combined Chilopoda and Diplopoda, and Nematoda, linear models). Reproduction mode was included only into the models of all taxa and of oribatids, as this data were limited in the other groups.

**Searching for core metazoan genes.** As a first-look into the functional capacities of the soil invertebrates in our study, we searched the genomes for potential loss of protein-coding genes. To make this analysis robust, we decided to focus on evolutionarily old genes that were present already in the last common ancestor of the animals. Using 11 species from across the Metazoa tree of life (Supplementary Data 7) which were part of the Orthologous MAtrix database (OMA[98]), we computed a list of 1482 core metazoan genes which were common to at least 9 of these 11 species using DCC (https://github.com/BIONF/dcc2) and pre-computed ortholog groups from the OMA DB. Given the evolutionary age of these genes and their conserved presence throughout the animal evolution, it appears likely that their loss has a

substantial functional impact. We preferred to use a custom core gene set over the standard BUSCO Metazoa ODB10 data set mainly for two reasons First, the BUSCO set with only 954 core genes is considerably smaller than the set computed by us. This gives us more power to detect differences in the presence/absence pattern of genes in the analysed taxa. Second, OMA groups represent cliques of orthologous proteins, i.e., all members within a group identify each other as pair-wise orthologs. As a consequence, OMA groups reconstruct orthologous relationships across proteins from many taxa with the highest precision among all available tools[98]. We then searched for orthologs of these 1482 metazoan core genes among the more complete (>50% BUSCO completeness) soil invertebrate genomes ($n = 177$) with fDOG-Assembly (https://github.com/BIONF/fDOG/tree/fdog_goes_assembly). fDOG-Assembly performs targeted, feature-aware ortholog search without the need for annotated genomes as the starting point. Due to the taxonomic breadth of our dataset, six separate ortholog searches were performed, each using the three most closely-related reference species with protein annotations available (Supplementary Data 8). Genes without orthologs in all investigated species were excluded from the following analyses. The resulting phylogenetic ortholog profiles were visualised with PhyloProfile (v1.8.6[99]) and clustered according to the euclidean distance of the presence and absence patterns of the ortholog groups. Hence, after visual inspection of the ortholog profiles, we were able to identify patches of core metazoan genes which were missing from certain groups.

We tested for gene ontology (GO) enrichment of the potentially missing genes using the InterProScan database[100] and the function runTest from the topGO package (v2.42.0[101]). For this GO-enrichment analysis, 1482 core metazoan genes were assigned to their ontology group(s), where GO annotation data were available. Using this list as a comparison, the two gene lists of interest (50 genes missing from the 78 Collembola species; 97 genes missing from the 54 Oribatida species) were separately tested for any significant enrichment of genes belonging to any of the three gene ontology groups (biological process, cellular component, or molecular function). Significant enrichment of a gene ontology term in the missing genes was stated when the category was represented by more than five genes in the list of 1482 core metazoan genes and with a significant over-representation in a Fisher's exact test ($p < 0.05$). Further, we manually screened putative functions associated with genes missing from Collembola and Oribatida assemblies in the UniProt database (accessed on 28.6.2023), aiming to identify functional or other relevant commonalities which might be missed by an algorithmic GO enrichment analysis.

The mean empirical probability of not being able to detect a particular gene in a taxon was 0.22 for all OMA genes, excluding those missing in springtails and oribatids. So this is also the probability of not finding a particular OMA gene in the genome of a new taxon. The probability that it is actually present in the majority of the taxa if it is also not found in the second sequenced species drops to 0.05; already in the third species in which the gene is not found, the probability that the gene is actually present in the majority of the species of the taxon is below the significance level.

**Statistics and reproducibility.** Genome analyses are based on Illumina genomes of 232 soil invertebrate species. Genome sizes could be estimated for 191 species. Metatranscripomic assignment was performed on 10 soil RNA samples. Structural equation models were fitted on genome properties of 143 taxa, including 27 myriapods, 34 oribatids, 68 springtails, 5 nematodes. Genomes of 177 species were assessed for the presence of core metazoan genes. Tests of normal distribution were performed to ensure that

assumptions of regression are fulfilled for the structural equation models.

**Reporting summary**. Further information on research design is available in the Nature Portfolio Reporting Summary linked to this article.

## Data availability

Vouchers are deposited in the collections of the Senckenberg Museum of Natural History Görlitz (SMNG), Germany. Raw sequence files and draft assemblies accessible through the ENA/NCBI project PRJNA758215. 28 S and COI barcodes are publicly available at dx.doi.org/10.5883/DS-TBGMI. Genome metadata can be accessed at the Genomes on a Tree (https://goat.genomehubs.org/projects/METAINVERT). Repeat elements can be accessed in the Dfam database (https://www.dfam.org/). Alignment of BUSCO genes and the resulting phylogenetic tree are available in FigShare (https://doi.org/10.6084/m9.figshare.24435052)[29]. Source data for Fig. 2 are part of Supplementary Data 1. Source data for Fig. 3 are provided in Supplementary Data 2. Source data for Fig. 5c, d are provided as Supplementary Data 4.

## Code availability

No custom code or mathematical algorithms are central for the conclusions of the paper. R commands for metatranscriptome analysis and structural equation models are deposited in FigShare[29]. A list of used software with versions are deposited in FigShare[29].

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

## Acknowledgements

This work is a result of the LOEWE Centre for Translational Biodiversity Genomics funded by the Hessen State Ministry of Higher Education, Research and the Arts (HMWK). This project contributes to the Soil Invertebrate Genome Initiative (https://tbg.senckenberg.de/sigi/), affiliated with the Earth BioGenome Project. Special thanks to Damian Baranski, Jürgen Otte and Jörg Müller for DNA extractions, to Astrid König for extraction and preparation of nematodes and tardigrades from Senckenberg cultures and soils, to Lena Bonassin and Jade Tessien for help with organising the datasets and barcodes, to Prof. Dr. Florian Grundler (INRES Molekulare Phytomedizin, Bonn University, Germany) for thousands of J2 juveniles of *Heterodera schachtii* and *Meloidogyne incognita* in ethanol from their cultures, and to Magnus Wolf for the BUSCO-to-phylogeny pipeline. Animal silhouettes originate from PhyloPic and they can be reused under Creative Commons licenses (http://www.phylopic.org).

## Author contributions

P.D., I.E., K.H., O.L., R.L., M.P., M.B. conceived and designed the experiments. P.D., K.H., H.M., R.L., performed the experiments. G.C., C.S., L.B., I.E., O.L., H.M., J.u.R., C.R., R.V., M.P., M.B. analysed the data. C.S., L.B., U.B., A.C., P.D., I.E., K.H., D.M., H.M., J.ö.R., J.u.R., C.R., R.S., A.S., K.T. contributed materials/analysis tools. G.C., M.P., M.B. wrote the paper.

## Funding

## Competing interests

The authors declare no competing interests.
