## [Peer Review File · Communications Biology]

This manuscript has been previously reviewed at another Nature Portfolio journal. This document only contains reviewer comments and rebuttal letters for versions considered at Communications Biology.

Reviewers' comments:

Reviewer #1 (Remarks to the Author):

In the submitted manuscript, the authors have satisfactorily addressed most of reviewers previous comments from previous submission to NatEco&Evo and in principle this paper would be ready for publication in its current form in Communications Biology. In my opinion, the changes in the structure and text, makes the manuscript much more clearer and the "resource" nature of the work is more evident.

I only have an extremely minor typo I found: Figure legend 3, it is still written as Figure 2, due to the old version of the figures.

Reviewer #2 (Remarks to the Author):

There were two main concerns in my initial review: (1) the parameters were not conservative enough and (2) the authors decided to make their own less conservative "core" gene set using ~1500 orthologs.

My sentiments are similar to Reviewer #3 in that the SEM results are sort of a hodgepodge and thus sort of a negative result, and consistent with multiple reviewers I feel that there really isn't a research question per se here -- but the authors have (re)focused on the resource nature of these genomes and it is a lot of work.

My only required comment relates to text on lines 246-248 given the less conservative gene set they used. Specifically I think they should remove the text: " which makes it likely that they are connected with the evolutionary emergence of the specific phenotypes of springtails and oribatids" for two reasons:

1. They have no data for oribatids since these genes seem to have no known function
2. The data and text below re: springtails and pyridine-containing compound metabolic processes is data-driven and a stronger support of this analysis than the speculation above.

BUSCO gets around this by using a much larger set of genomes than the 11 considered here -- which is why the list is ~1000 not 1500 -- but I think removing the text above and focusing on what their data show is a nice use case of the resource genomes.

To be clear I agree that since the genes are missing in all of their genomes they probably were lost; my concern is with the "core"ness of the additional 500 genes. The bacterial literature has lots of work on this since parameters do matter a lot in the bioinformatics, but given the later bits of that paragraph I don't think it matters as much and is a strength, namely a lot of genes in non-models are not well characterized.

Reviewer #3 (Remarks to the Author):

The authors did a very thorough revision to address the comments of the four referees. The paper is overall very well written and the analyses are of a very high standard throughout. The study breaks new ground in providing (partial!) genome sequences for many hitherto unstudied deep lineages of invertebrates. Yet, as all of the reviewers, I remain uneasy about the design of the study. It shows a few interesting points but none of them are conclusive. Specifically: the phylogeny is only very crudely annotated, as little information is provided on the taxa on this tree beyond the phylum level. The altitudinal gradient is described in simplistic terms and not further explored. The genome size study is no more than a first pass through these groups, but too crude for any mechanistic understanding. The study of population genomics is a single type of analysis within a particular software. Hence, the study simply presents a few examples of what can be done with this kind of data, yet does not really add truly new (citable) information on the various topics of ecology, genome evolution and population genomics. Perhaps as a minimum requirement, the phylogenetic tree should be examined in greater detail and properly documented so that the reader has a better understanding of the phylogenetic breadth covered (*within* each of the phyla) and also can reuse the tree inferred here and the alignment from which it was generated.

Reviewers' comments:

Reviewer #1 (Remarks to the Author):

In the submitted manuscript, the authors have satisfactorily addressed most of reviewers previous comments from previous submission to NatEco&Evo and in principle this paper would be ready for publication in its current form in Communications Biology. In my opinion, the changes in the structure and text, makes the manuscript much more clearer and the "resource" nature of the work is more evident.

Thank you!

I only have an extremely minor typo I found: Figure legend 3, it is still written as Figure 2, due to the old version of the figures.

Line 136: Thank you, it is corrected.

Reviewer #2 (Remarks to the Author):

There were two main concerns in my initial review: (1) the parameters were not conservative enough and (2) the authors decided to make their own less conservative "core" gene set using ~1500 orthologs.

My sentiments are similar to Reviewer #3 in that the SEM results are sort of a hodgepodge and thus sort of a negative result, and consistent with multiple reviewers I feel that there really isn't a research question per se here -- but the authors have (re)focused on the resource nature of these genomes and it is a lot of work.

My only required comment relates to text on lines 246-248 given the less conservative gene set they used. Specifically I think they should remove the text: " which makes it likely that they are connected with the evolutionary emergence of the specific phenotypes of springtails and oribatids" for two reasons:

1. They have no data for oribatids since these genes seem to have no known function
2. The data and text below re: springtails and pyridine-containing compound metabolic processes is data-driven and a stronger support of this analysis than the speculation above.

Line 258: This sentence part is now removed.

BUSCO gets around this by using a much larger set of genomes than the 11 considered here -- which is why the list is ~1000 not 1500 -- but I think removing the text above and focusing on what their data show is a nice use case of the resource genomes.

Thank you!

To be clear I agree that since the genes are missing in all of their genomes they probably were lost; my concern is with the "core"ness of the additional 500 genes. The bacterial literature has lots of work on this since parameters do matter a lot in the bioinformatics, but given the later bits of that paragraph I don't think it matters as much and is a strength, namely a lot of genes in non-models are not well characterized.

Reviewer #3 (Remarks to the Author):

The authors did a very thorough revision to address the comments of the four referees. The paper is overall very well written and the analyses are of a very high standard throughout. The study breaks new ground in providing (partial!) genome sequences for many hitherto unstudied deep lineages of invertebrates.

Thank you, it was indeed a major work to generate these genomes.

Yet, as all of the reviewers, I remain uneasy about the design of the study. It shows a few interesting points but none of them are conclusive. Specifically: the phylogeny is only very crudely annotated, as little information is provided on the taxa on this tree beyond the phylum level. The altitudinal gradient is described in simplistic terms and not further explored. The genome size study is no more than a first pass through these groups, but too crude for any mechanistic understanding. The study of population genomics is a single type of analysis within a particular software. Hence, the study simply presents a few examples of what can be done with this kind of data, yet does not really add truly new (citable) information on the various topics of ecology, genome evolution and population genomics. Perhaps as a minimum requirement, the phylogenetic tree should be examined in greater detail and properly documented so that the reader has a better understanding of the phylogenetic breadth covered (*within* each of the phyla) and also can reuse the tree inferred here and the alignment from which it was generated.

We now provide a more detailed version of the tree as a supplementary figure. Families are annotated on this tree, to better show the phylogenetic breath covered. We also provide the text tree file and the alignment for potential reuse on FigShare (doi: 10.6084/m9.figshare.24435052).